# Applying Density-Based Clustering for the Analysis of Emission Events in Real Driving Emissions Calibration

Sascha Krysmon [1],*, Stefan Pischinger [1], Johannes Claßen [1], Georgi Trendafilov [1], Marc Düzgün [1], Frank Dorscheidt [1], Martin Nijs [2] and Michael Görgen [2]

[1] Chair of Thermodynamics of Mobile Energy Conversion Systems, 52074 Aachen, Germany
[2] FEV Europe GmbH, 52078 Aachen, Germany
* Correspondence: krysmon@tme.rwth-aachen.de; Tel.: +49-241-80-48078

**Abstract:** Further reducing greenhouse gas and pollutant emissions from road vehicles is a major task for the automotive industry. Stricter regulations regarding emissions and fleet fuel consumption require the continuous development of new powertrains and methods. In particular, the combination of hybrid powertrains on the technical side and the focus on real driving emissions (RDE) on the legislative side pose significant challenges to the vehicle calibration process. Against this background, new test methods and environments are being investigated to counteract the high number of interactions between hybrid drive systems and quasi-infinite test conditions due to RDE. Complementary to new test environments, innovative methods for data analysis are needed that allow the exploitation of the complete potential of measurement data. The application of such a method in the field of emission calibration is presented in this paper. For this purpose, a clustering method (HDBSCAN) is applied to critical sequences from emission tests. Within this presentation, the clustering process is based on a single signal only. This paper shows how signals of various characteristics can be processed with dynamic time warping and generically structured with the clustering method used. Here, 959 single events are automatically categorized into 24 clusters. This provides a new basis for system evaluation, enabling the automatic identification, categorization, and prioritization of calibration weaknesses. Using twelve signals of different characteristics, the generic usability of the clustering method is demonstrated.

**Keywords:** real driving emissions; emission calibration; virtual calibration; data analysis; clustering; density-based clustering; HDBSCAN

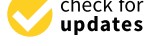



## 1. Introduction

In modern vehicle and powertrain design, climate protection and emission reduction are high priorities [1]. The trend and direction of vehicle development are influenced by a reduction in energy and fuel consumption, as well as the optimization of the overall system with the aim of producing zero-impact emission vehicles [2,3]. Furthermore, legislation continues to drive this change by adjusting emission regulations [4,5].

With the central component of the EU6d legislation [6], automobile testing under real driving conditions is mandated for the first time. The EU6d standards limit the test scenarios with regard to acceleration intensities, route characteristics, and environmental factors such as temperature and altitude. However, the potential scenarios for real driving emissions (RDE) generate an almost infinite range of testable space that must be taken into account, tested, and validated. The current planning for EU7 legislation [7] suggests that there will be further expansion [8]. With a heightened emphasis on real-world testing and the potential elimination of limitations on the dynamics criteria, the testing area may expand even further, thus presenting greater challenges to available testing resources.

With the increasing complexity resulting from the electrification and hybridization of vehicles, the importance of testing and analyzing measurement data using suitable

methods is increasing. The rise in the number of components, such as electric machines and high-voltage batteries, introduces additional interdependencies into test matrices. When assessing the robustness and quality of the complete powertrain, it is essential to consider all possible worst-case scenario combinations. Novel techniques are necessary to reduce the number of required tests and ensure the quality and statistical robustness of the calibrations. Thus, this paper proposes a method to improve the data analysis and statistical evaluation of vehicle calibration. To facilitate its application in emission calibration, this approach clusters critical events based on signal traces. A single signal is utilized solely for clustering in this demonstration, while, in general, the approach could be extended to consider a multitude of signals to describe the present data on a more detailed level.

The presentation begins with a discussion of the current state-of-the-art methods in the calibration process to justify the necessity for a novel approach to data analysis. This is followed by a brief overview of the overall methodology. Subsequently, the utilized data source and fundamental steps of the clustering procedure are presented. The impact of dynamic time warping on the present data is first discussed in the results section. Next, the HDBSCAN (Hierarchical Density-based Spatial Clustering of Applications with Noise) algorithm results regarding emission event traces are presented in two parts. Initially, a section of the data is manually transferred into reference clusters, and the HDBSCAN outcomes are compared in terms of the Adjusted Rand Index ($ARI$) for various signals. Following, the HDBSCAN is applied to the complete set of events using only the engine speed signal. Finally, this paper examines the utility of clustering for vehicle calibration purposes using the presented results as an example.

*State-of-the-Art—Novel Methods for Vehicle Calibration*

Addressing the challenges posed by system complexity and test condition boundaries can be achieved with modifications to the testing facility or test scenario generation methods. Enhanced test facilities can increase the speed of required tests and improve test-to-test reproducibility, while innovative methods for test scenario generation can reduce the overall number of necessary tests by focusing on vehicle-specific relevant aspects. Therefore, this excerpt of the state-of-the-art focuses on advanced testing facilities and the methods used for data evaluation and test scenario generation. While current research emphasizes improving test bed facilities, limited research has been published regarding dedicated data analysis in the context of vehicle calibration.

Virtual environments and intelligent test scenario design methods are central to investigations on vehicle calibration, as shown in [9]. In vehicle development, calibration tasks commonly use Model-in-the-Loop (MiL), Hardware-in-the-Loop (HiL) [10,11], and Engine-in-the-Loop (EiL) [12–14]. These X-in-the-Loop (XiL) test benches are increasingly used for testing real driving emissions and applications, as noted in numerous publication sources [15–18]. One advantage of these test benches is their high degree of flexibility. Pre-conditioning time can be reduced by simulating component temperatures and ambient conditions, although this requires adequate models for the relevant conditions.

Examples of calibration on HiL test bench environments are discussed and presented for various use cases. Although the topic of emission calibration is discussed for early RDE calibration [11,19,20] in conventional vehicles, the increased complexity of the powertrain motivates its application in hybrid powertrains [21–23]. In [24], the focus on virtual drivability calibration on an EiL test bench is discussed. High correlation is shown for the detailed simulation models for drivability and transmission with low deviations in emissions compared with chassis dynamometer tests. The use of EiL can lead to a reduction in calibration efforts by up to 30% and costs by up to 20%. Additionally, the used models are suitable to enable potential objectification in drivability [25]. Furthermore, Schmidt et al. offer an extensive overview of system validation methods for drivability in [26].

In parallel with the use of innovative test benches and simulation techniques, multiple approaches are being developed to pivot testing efforts toward use-case-relevant scenarios. These approaches can be categorized into four groups (generic test cycles, real-world routes,

real-world driving behavior, and worst-case estimations) as described in [8]. Generic test cycles (e.g., ADAC BAB cycle or RTS95) are commonly used fleet-generically on chassis dynamometers and offer a high level of comparability and a low vehicle-specific effort. Real-world route retracing with operating point reproduction [27–30] can transfer real-world driving behaviors or environments to test benches or simulations, with the added ability to test different traffic scenarios [15]. Synthetically generated statistical driving profiles—primarily using Markov chains [31]—are utilized to represent regional driving behavior and are the focus of the research in [32–37]. Worst-case cycles generated using Design-of-Experiments (DoE) based on engine test benches or simulations are used for the final testing of the most intensive test cases, suggesting a safety-oriented approach, as described in [38–41].

Data analysis methods are used to support modeling, calibration, and defining relevant test scenarios in addition to enabling XiL test benches [42]. For instance, Isermann et al. outline an approach utilizing optimization algorithms in an offline simulation for base calibration focusing on emissions and fuel consumption [43]. Wasserburger et al. propose a methodology in [44,45] for generating test cycles from engine-operating points and using these as input values for an optimization algorithm that adjusts the calibration of specific functions to optimize vehicle emissions. Moreover, offline powertrain models use nearest neighbor clustering algorithms to frontload the engine base calibration in [46], and a methodology for model-based smooth calibration is presented in [47]. The investigation of neural networks is the main topic in [48] for developing models for the optimization of baseline calibration. Steinbach et al. analyze the virtualization of emission calibration and emission modeling for RDE optimization in [49]. In [50], a methodology for emission simulation is developed, which is further discussed for use on EU7 applications in [51]. Further research and publications within the context of advanced data analysis in automotive development include the clustering of vehicle trajectories [52] and calibration of autonomous driving systems in the automotive sector [53,54].

To fully utilize of the advantages resulting from the design of test scenarios and virtual test beds, a targeted analysis of measurement data is highly relevant. In the context of emission calibration, recognizing patterns, trends, and clusters is a primary challenge when assessing the quantitative effects of the identified weak spots in RDE applications. Such approaches remain largely unexplored in the current state of the research.

## 2. Materials and Methods

The presented methodology is part of a continuous validation concept. The application of clustering refers to critical sequences from emission measurements, which are referred to as events; the definition and detection of these events are presented in the following. The necessary pre-processing of the events as well as the methodology for the formation of the clusters in the HDBSCAN procedure are explained. Finally, necessary evaluation criteria (Silhouette Score, Density-Based Cluster Validation, and Adjusted Rand Index) are presented.

The overall methodology for measurement-based RDE validation is presented and discussed in detail in [55] and includes—as shown in Figure 1—four topics:

1. Event-based RDE validation using multiple test environments.
2. Identification of calibration potentials.
3. Quantification of statistical safety.
4. Dynamic and predictive cycle generation.

The clustering application is part of topic 2, which focuses on the identification of optimization potentials in vehicle calibration. Topic 3 deals with the evaluation of the statistical reliability of the used measurement database. In topic 4, the creation of test scenarios based on [56] is implemented.

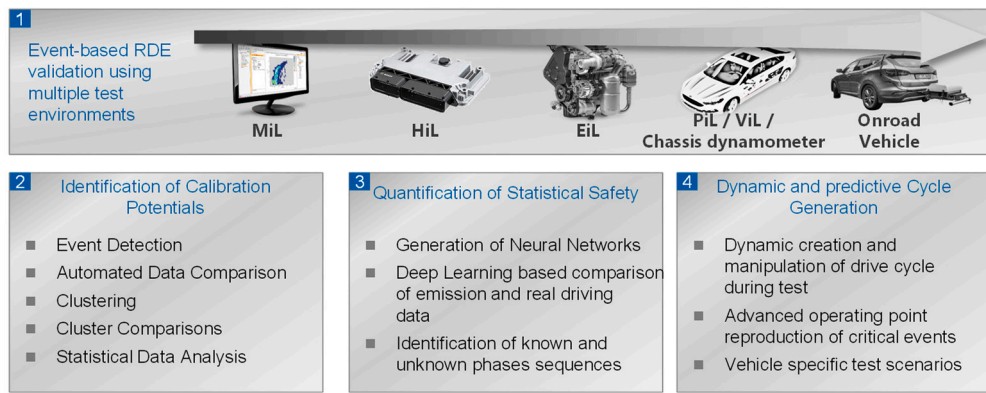

**Figure 1.** Concept of the RDE validation methodology as context to the clustering process.

## 2.1. Data Source

Emission measurements from a vehicle calibration project serve as the data basis for the clustering process. The data originate from an RDE validation campaign of a production vehicle (Table 1) with a series production engine control unit (ECU) dataset.

**Table 1.** Test vehicle specification.

| Characteristic | Unit | Value |
|---|---|---|
| Vehicle weight | kg | >2000 |
| Fuel | — | Gasoline |
| Engine type | — | Turbo-charged 8 cylinder |
| Engine power and torque | kW/Nm | >400/>600 |
| Cubic capacity | $cm^3$ | ~4000 |
| Transmission | — | Automatic Transmission (AT) |
| Drivetrain | — | All-wheel drive (AWD) |
| Exhaust aftertreatment system (EATS) | — | Three-way catalytic converter (TWC) and gasoline particulate filter (GPF) |
| Condition of EATS | — | Stabilized EATS (~70% ) and aged EATS (~30% ) |
| Emission target | — | EU6d |

The total of 78 measurements was first analyzed for critical $NO_X$ emission intensities using event detection, as described in Section 2.2, and transformed into 959 events. The measurements carried out include temperatures between $-7\,°C$ and $35\,°C$. In addition to WLTC measurements, 7 different RDE speed profiles were tested on an emission chassis dynamometer test bench. Furthermore, 2 different routes in different drive modes were tested with a portable emission measurement system (PEMS) on-road. The tests were carried out with stabilized exhaust aftertreatment systems (>3000 km , ~70% of the driven tests) as well as aged exhaust aftertreatment systems (~30% of the driven tests). All data were resampled to a 1 Hz frequency prior to the event detection as most of the emission measurements are only available in this resolution.

## 2.2. Events and Event Detection

An event denotes a time sequence of increased emission intensities from emission test measurements. It encompasses all signals recorded within the sequence, including ECU measurement and test bench measurement data. To detect these sequences, the emission measurements are scanned automatically using a moving integrating window that assesses the distance-specific emission intensity, as described in [57]. Events typically show durations of 8 s to 120 s, though the duration is variable and dependent upon vehicle emissions. The detection methodology is not the scope of this paper and can be seen in detail in [57]. The applied thresholds used for the data here are listed in Table 2.

**Table 2.** Threshold data for event detection.

| Signal | Unit | Urban | Rural | Motorway |
|--------|------|-------|-------|----------|
| Speed breakpoints | km/h | 0 | 80 | 160 |
| Intensity breakpoints | mg/km | 30 | 30 | 30 |

### 2.3. Pre-Processing of Events and Distance Calculation

For the further processing of the events into clusters, a comparison of these is necessary. To accomplish this, a distance matrix is created, which serves as the foundation for the clustering process. The distance matrix is $n \times n$ with dimensions that correspond to the number of events it contains. The distance matrix contains the pairwise distance between each event toward each of the other events. Due to varying event durations, a basic Euclidean comparison is not feasible. Therefore, a dynamic time warping (DTW) approach is used. Figure 2 presents a direct comparison (left), a prior signal synchronization using a best-match approach (center), and the use of DTW (right).

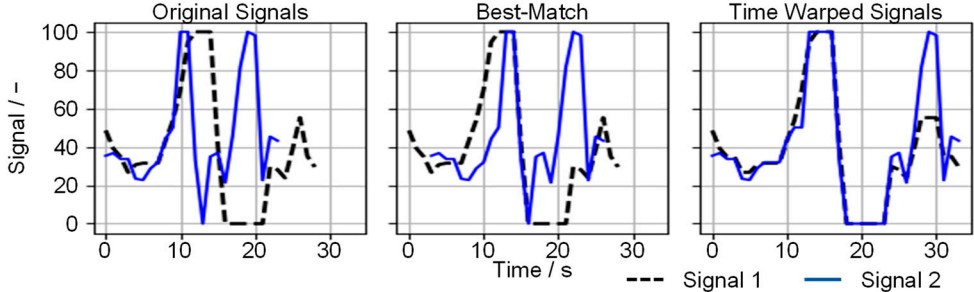

**Figure 2.** Comparison using original, best match synchronized, and time-warped signals.

While the best match approach can decrease the distance between two signals by synchronizing them beforehand, it cannot include slight differences and offsets. Additionally, when signals have various lengths, not all data points can be considered. The DTW adjusts both signals to each other to minimize the distance between both traces by duplicating individual points [58,59].

$$d_{L_n}(x,y) = \left( \sum_{i=1}^{\delta} (x_i - y_i)^n \right)^{\frac{1}{n}} \tag{1}$$

After pre-processing the signals with DTW, both signals share a standardized duration. Then, the signals are evaluated with a distance calculation using $L_n$ norm (Equation (1)) [58]. For each sample $i$ of signals with total duration $\delta$ (according to DTW), the difference between the two signals $x$ and $y$ is calculated. Here, n = 2 is used for the application of the Euclidean distance between the DTW processed traces.

The effects of DTW are examined in Section 3.1, as manipulating signal traces, though offering advantages for dynamic signal comparisons, increases the likelihood of overfitting signals. This can result in falsely indicating high correlations between traces.

While this application only considers comparisons of events using a single signal at a time, the clustering approach could incorporate a multitude of signals by modifying the distance matrix. When comparing multiple signals, the dimension of the required distance matrix for the clustering algorithm can be kept constant by calculating the distance according to Equation (2).

$$d_{L_n}(x,y,\,s) = \left( \sum_{s=1}^{|s|} c_{x,y,s} \cdot d_{L_n,s}(x,y)^2 \right)^{\frac{1}{2}} \tag{2}$$

While each signal $s$ in the signal set $S$ is first processed individually according to Equation (1), resulting in a scalar distance $d_{L_n,s}$, the overall distance in the multidimensional space can be calculated using the $L_2$ norm. A weighting factor ($c_{x,y,s}$) is introduced to

balance the weight of different signals and to potentially increase the weight of comparisons with a high distance of events for a specific signal.

## 2.4. Clustering Method

The hierarchical density-based method HDBSCAN [60] is used to cluster the data based on the distance matrix. This approach does not require information on either the desired number of clusters or the maximum distances between them. The variability in signal types with different dynamic behaviors and magnitudes is a critical aspect of an appropriate clustering method in emission calibration. Various clustering algorithms could be explored for application on calibration data (such as hierarchical, partitioning, Fourier transformation-based optical clustering, etc.). However, previous analyses indicate that the HDBSCAN method has promising features [61]. A comparison of hierarchical, partitioning, and density-based clustering methods is detailed and evaluated in [61].

Within the HDBSCAN method, hierarchical clustering [62–64] is utilized initially. In contrast to the conventional method, there is no requirement to specify a maximum distance [65] in the cluster, as the optimization algorithm determines it individually for each part of the cluster tree—similar to partitioning methods [66–68]. Furthermore, the HDBSCAN algorithm can define signals that decrease overall cluster quality as outliers.

When applying the algorithm, three parameters are used to control the clustering algorithm:

- Minimum cluster size $C_{minSize}$.
- Minimum density $\rho_{min}$.
- Minimum distance between two clusters $\varepsilon_{min}$.

The analyses in [61] demonstrate the feasibility of defining $C_{minSize}$ and $\rho_{min}$ identically. Additionally, the Leaf method in HDBSCAN is applied here, as it displays favorable outcomes in [61] by tending toward generating multiple smaller more closely linked clusters. If the cluster contents indicate the same phenomena, the engineer may manually merge them post-automatic clustering.

The outcome of one HDBSCAN clustering execution on a single signal is defined as a cluster set in this paper. Different cluster sets can result from using distinct signals or event input data to the HDBSCAN execution.

## 2.5. Characteristic Values for Cluster Evaluation

The validation metrics Silhouette Score, Adjusted Rand Index ($ARI$), and Density-Based Cluster Validation ($DBCV$) are used to evaluate the cluster results.

The Silhouette Score is an intensive measure for evaluating clusters [69]. To determine the Silhouette Score, the similarities within a cluster and differences with the other clusters are evaluated. For each element $o$ of the elements $O_{C_i}$ in a cluster $C_i$, the average distance (Equation (1)) to all elements in the cluster is determined. The result is described by $a(o)$. In addition, the average distance to all elements of the nearest cluster $b(o)$ is identified. Using Equation (3), the silhouette $s(o)$ is calculated. The Silhouette Score of a cluster $S(C_i)$ is then defined according to Equation (4) [69]

$$s(o) = \frac{b(o) - a(o)}{\max\left(a(o),\, b(o)\right)} \tag{3}$$

$$S(C_i) = \frac{1}{\left|O_{C_i}\right|} \cdot \sum\nolimits_{o=1}^{\left|O_{C_i}\right|} s(o) \tag{4}$$

Silhouette Scores result in values between $-1$ and 1. Results close to 1 indicate very well-separated and dense clusters, $-1$ reveals a cluster misattribution [69].

The $DBCV$ (Equation (5)) is an intrinsic evaluation measure, which evaluates clusters of any shape based on their density among each other [70,71]. It compares the density within clusters to the density between clusters with resulting values between $-1$ and 1. Negative $DBCV$ values indicate clusters that have a density lower than the environment.

For a set of clusters $C$ containing $|C|$ clusters, the size of the cluster $C_i$ in relation to the number of all considered elements $O$ is factorized by the cluster validity $V_C(C_i)$ [70].

$$DBCV(C) = \sum_{i=1}^{|C|} \frac{|C_i|}{|O|} \cdot V_C(C_i) \tag{5}$$

*ARI* is an extensive measure for the evaluation of correct cluster assignment to their expected value (Equation (7)). It evaluates the correct assignment of elements to a cluster when the correct assignment of these elements is already known. The *ARI* is based on the Rand Index *RI* (Equation (6)), which is formed by the ratio of the sum of True Positive *TP* and True Negative *TN* values and the total of *TP*, *TN*, False Positive *FP*, and False Negative *FN* values. For the calculation of the *ARI*, the *RI* is corrected by the expected value $E(RI)$ of a random assignment. The *ARI* has an expected value of 0 and a maximum value of 1. $ARI = 1$ corresponds to ideal agreement [72,73].

$$RI = \frac{TP + TN}{TP + FP + TN + FN} = \frac{TP + TN}{\binom{n}{2}} \tag{6}$$

$$ARI = \frac{RI - E(RI)}{\max(RI) - E(RI)} \tag{7}$$

For the application of *ARI* to the emission events in this paper, a prior manual assignment of the sampled events into clusters is necessary. Subsets are randomly selected for 12 different signals from the event database and manually sorted into reference clusters. This process is performed on a purely visual basis. Table 3 shows an overview of the signals, the number of events, and the manually created reference clusters.

**Table 3.** List of analyzed signals and number of events for manual definition of reference clusters.

| Signal | Events Used | Reference Clusters Defined |
|---|---|---|
| Engine speed | 366 | 7 |
| Vehicle speed | 448 | 29 |
| Flag fuel cut-off | 648 | 16 |
| Relative air charge | 304 | 18 |
| Pedal position | 271 | 23 |
| Voltage of two-point downstream lambda sensor | 725 | 10 |
| Engine torque | 494 | 43 |
| Relative fuel mass | 461 | 36 |
| Catalytic converter temperature | 885 | 11 |
| Exhaust gas mass flow | 174 | 12 |
| Actual ignition angle | 237 | 15 |
| Optimal ignition angle | 561 | 30 |

The manually created reference is compared with the automatically created cluster sets in Section 3.2. The signals are selected in a way to use signals of different characteristics for the validation. The signals include very dynamic characteristics with a high range of values (e.g., engine speed), but also signals with rather smooth dynamics (e.g., vehicle speed or catalytic converter temperature). In addition, signals that change abruptly (e.g., the voltage of the two-point downstream lambda sensor) and binary signals (e.g., the flag fuel cut-off) are used. The choice of signals is exemplary and allows for verification of the generic application of the method to all available measurement data.

## 3. Results

In this section, the results from applying the clustering process to emission measurements are presented. The structure of the section is displayed in Figure 3.

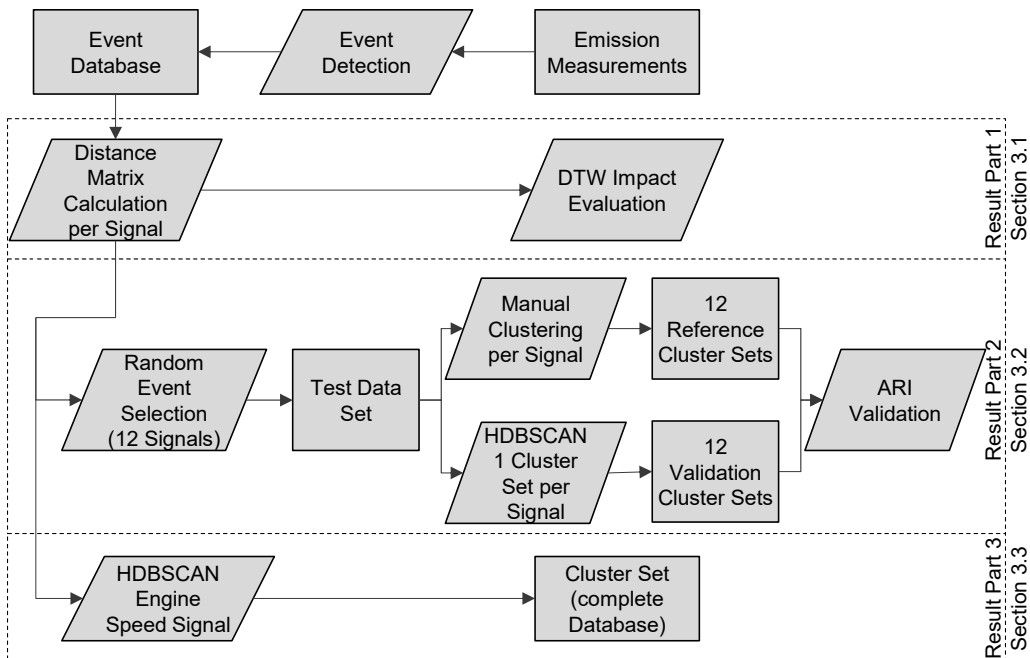

**Figure 3.** Schematic overview of the cluster application.

First, adjustments from data pre-processing are shown. Then, a validation of the HDBSCAN application on the used data is discussed for different signals, evaluating the impact of the characteristic values for cluster evaluation. Subsequently, the exemplary application of the clustering procedure to the entire data is presented.

### 3.1. Pre-Processing of Data

The DTW correction to the signals enables the compensation of offsets and smaller differences. This simplifies the comparison of signals of different durations and characteristics, but, at the same time, distorts the signals. Partially significant profile sections may align, resulting in the incorrect classification of two events as similar. The impact of DTW results is evaluated visually for randomly selected comparisons. Figures 4 and 5 display the impact using the downstream lambda sensor voltage $U_{\text{HEGO}}$ and the vehicle speed $v$.

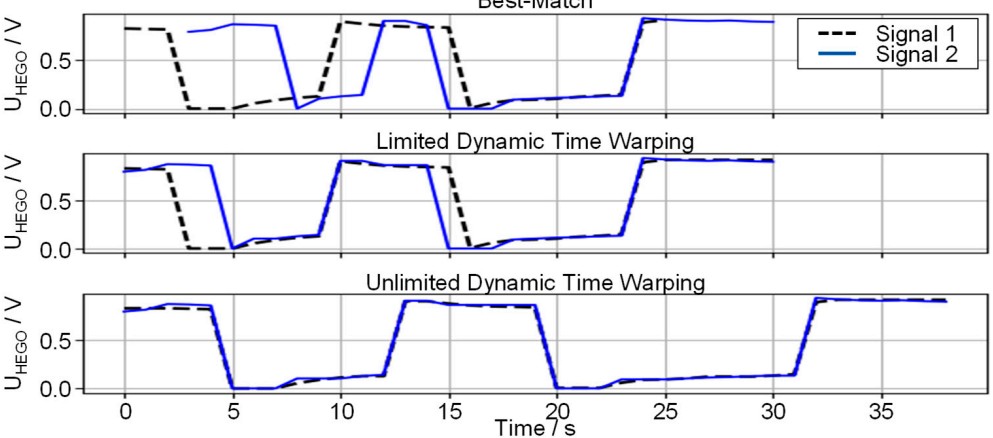

**Figure 4.** Impact of limited and unlimited DTW based on the downstream lambda sensor voltage.

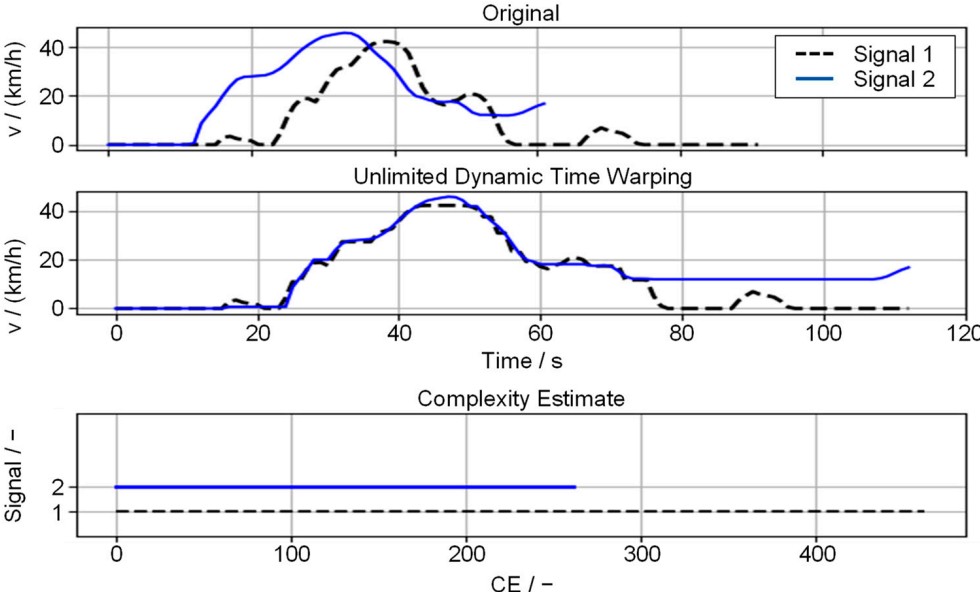

**Figure 5.** Application of complexity estimate correction to DTW comparisons.

The difference between limited and unlimited DTW is illustrated in Figure 4. The curves of the reference profile (signal 1) and the event to be compared (signal 2) show similar characteristics. The curves start with a voltage of $U_{\text{HEGO}} = 800$ mV, indicating a normal state of the catalytic converter in the $\lambda = 1$ (stoichiometric mixture) operation. After $\Delta t = 5$ s, the sensor voltage drops to $U_{\text{HEGO}} = 0$ mV–100 mV. This indicates an oversaturation of the catalytic converter with oxygen. This results from a fuel cut-off maneuver. Here, the absence of fuel injection results in air being pumped through the engine and exhaust aftertreatment system (EATS). The oxygen contained in the air is stored in the catalytic converter until the maximum storable amount of oxygen is reached. The remaining oxygen is then detected by the sensor. After a further $\Delta t = 8$ s, the fuel cut-off is terminated and fuel is re-injected. However, in the already oversaturated state of the catalytic converter, $NO_X$ emissions cannot be reduced. As a consequence, the ECU enriches the mixture to operate at $\lambda < 1$ to purge the oxygen from the catalytic converter due to the oxidation of CO and HC ($\Delta t \cong 10$ s with voltage regaining $U_{\text{HEGO}} \cong 800$ mV). This maneuver is repeated in both events. Signal 1 and signal 2 are to be considered equal from a technical point of view. The slight difference in the durations of the enriched and lean phases has a minor influence on the resulting emission event. Here, an unlimited DTW is useful.

A similar behavior is seen for further signals and events. However, an overfitting of signals using DTW must be prevented. For this purpose, the complexity of a signal is incorporated into the distance measure using the complexity estimation presented by Batista et al. in [74]. The distance measure *EDTW* for the similarity of two events $x$ and $y$ is expressed as the product of the distance $d_{\text{DTW}}$ (calculation according to Equation (1)) and a correction factor *CF* (Equation (10)). The correction factor describes the ratio of the complexity values *CE* of the signals (Equation (9)). The complexity *CE* is a measure used to judge changes in the signal course according to Equation (8). It is calculated by identifying the distances of consecutive signals values [74].

$$CE(x) = \sqrt{\sum_{i=1}^{|x|-1}(x_i - x_{i+1})^2} \tag{8}$$

$$CF(x,y) = \frac{\max(CE(x), CE(y))}{\min(CE(x), CE(x))} \tag{9}$$

$$EDTW(x,y) = d_{\text{DTW}}(x,y) \cdot CF(x,y) \tag{10}$$

The effect of the correction is shown in Figure 5. The velocity traces of two events are first adjusted with unlimited DTW. Since the signal of the velocity of the second event $v_{E2}$ (blue) is clearly shorter than signal one, it is distorted stronger. Similarly, the course is less complex. The difference in the complexities in the lower plot shows the quantitative evaluation, which leads to a correction factor of $CF(v_{E1}, v_{E2}) = 1.92$ with $CE(v_{E1}) = 483$ and $CE(v_{E2}) = 252$.

In this way, different complexities and event durations are considered when applying the DTW correction. Thus, an overfitting due to DTW cannot be prohibited but will at least be considered in the distance matrix by the correction factor. The further creation of the distance matrix for application to the HDBSCAN clustering method is thus calculated using the complexity correction-dynamic time warped distances.

### 3.2. Verification of HDBSCAN Using Data Extract

The HDBSCAN procedure is initially applied per signal to the selection of emission measurement data shown in for verification. Accordingly, 12 cluster sets are identified. The assignment of the events to the categories per cluster set can differ. The combination of different signals and agreement on the division of the same event groups into the same clusters in various cluster sets are not evaluated here.

The results of applying HDBSCAN to the test data are shown in Table 4. While the agreement with the test clusters represented by *ARI* is predominantly good, the *DBCV* evaluates the compactness of the clusters independently of the previously manual assignment. With values of $DBCV > 0.5$ and a mean value over all signals of $\overline{DBCV} = 0.57$, a good compactness of the data is interpreted here.

**Table 4.** Validation of HDBSCAN clustering results based on ARI and DBCV.

| Signal | Identified Clusters | Outliers | *ARI* | *DBCV* |
|---|---|---|---|---|
| Engine speed | 8 | 20 | 0.68 | 0.57 |
| Vehicle speed | 10 | 8 | 0.85 | 0.63 |
| Flag fuel cut-off | 61 | 24 | 0.7 | 1.00 |
| Relative air charge | 22 | 5 | 0.99 | 0.52 |
| Pedal position | 22 | 16 | 0.71 | 0.47 |
| Voltage of two-point downstream lambda sensor | 19 | 27 | 0.62 | 0.49 |
| Engine torque | 27 | 36 | 0.78 | 0.50 |
| Relative fuel mass | 28 | 16 | 0.82 | 0.44 |
| Catalytic converter temperature | 11 | 1 | 0.50 | 0.68 |
| Exhaust gas mass flow | 14 | 3 | 0.71 | 0.62 |
| Actual ignition angle | 17 | 19 | 0.80 | 0.48 |
| Optimal ignition angle | 22 | 43 | 0.69 | 0.40 |

The mostly low number of classified outliers using HDBSCAN show that the density-based clustering provides good results here as well. High numbers of outliers are only observed for the signals of the optimum ignition angle and the engine torque. These two signals exhibit very dynamic trajectories, which provide high requirements for DTW and clustering due to the different signal lengths and complexities.

Examples of clusters are shown in Figure 6. The original signals of the events assigned to a cluster are shown in gray without applying DTW and without specific best-fit synchronization. The blue trace shows the center event of the cluster generated using the bary-center approach [75]. The bary-center is a synthetic trajectory computed with DTW that compensates for local temporal shifts in the signal trajectories. This allows to summarize the cluster in a representative form so that the analysis of the profile trajectories is simplified. Thus, the information in a cluster can be described by the representative signal and the cluster size.

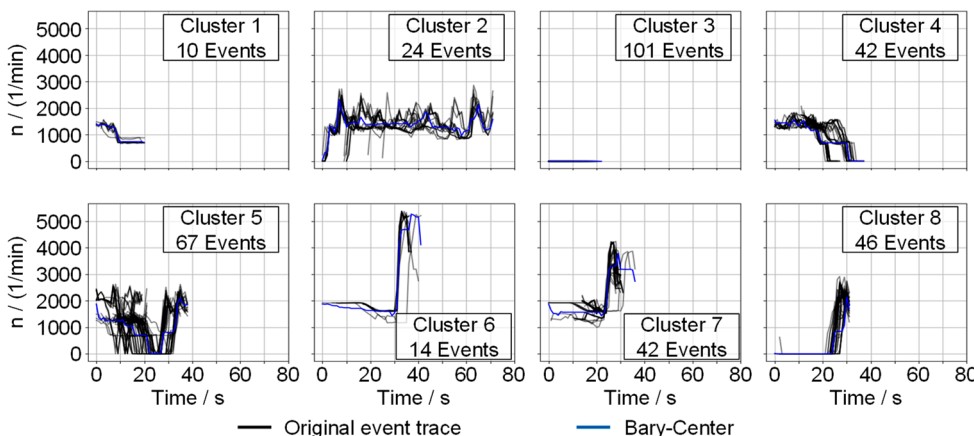

**Figure 6.** Extract of HDBSCAN-created clusters for engine speed.

The distribution of the clusters shows a reasonable categorization regarding the profile and length of the events. While rather short events are contained in clusters 1 and 3, cluster 2 shows the group of the longest events. The other clusters show similar event durations.

Cluster 3 shows an event where the engine is off during the event. This phenomenon is not based on a technical aspect of the engine or EATS but rather caused by the event detection procedure itself. Due to the window-based integration approach, each sample is assigned a distance-specific emission intensity that includes previous and following measurement samples. This method compensates for smaller synchronization errors by calculating the distance-specific intensity after filtering driven vehicle speed and emitted emissions. A direct sample-wise calculation has the disadvantage of being highly dependent on a high synchronization quality and resolution of the measurement device [57]. However, due to the filtering, the samples where no emissions are created can be assigned with the intensity resulting from earlier or later phases. Such phenomena cause the displayed cluster 3. The cluster 3 sequences are such sequences where the engine is off and no emissions are produced. Due to the low traveled distance during vehicle standstill, the braking or acceleration phases prior to or post the identified sequences lead to a high distance-specific emission level for the standstill phases. While the later or earlier samples (after drive-off or before standstill) are not considered as critical due to the higher traveled distance, the standstill phases are considered as such.

### 3.3. Application of HDBSCAN on the Complete Dataset

After applying the clustering algorithm to a partial extract of the total data, the algorithm is applied to the entirety of the 959 $NO_X$ events. As a reference signal, the engine speed is used. An automated definition of the minimum cluster size $C_{minSize}$ and $\varepsilon_{min}$ is used. The definition is performed by calculating the *DBCV* for cluster sets from $C_{minSize} = 3$ to 16. The setup that reaches the maximum *DBCV* is used. $\rho_{min}$ is defined as identical to $C_{minSize}$. Based on this, $\varepsilon_{min}$ is then iterated and, consistently, the value reaching the highest *DBCV* is selected.

When applied to the total amount of data, the data variety increases. This causes a large number of events to be classified as outliers, in contrast to the further results of the verification run. To overcome the high amount of outliers, an iterative re-clustering approach is implemented, as shown in Figure 7.

Initially, the overall event database is used for the first loop. For the second loop, the outliers of a cluster set are defined as new input data and subjected to re-clustering. This is repeated until the share of detected outliers in the total dataset reaches a maximum of 10%. The resulting cluster sets of each loop are appended to the previous iterations.

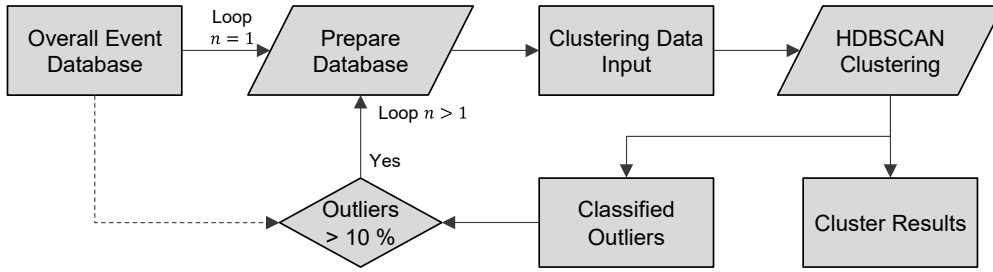

**Figure 7.** Procedure of outlier re-clustering.

The results of this iterative process are shown in Table 5. After four iterations, the number of events classified as outliers (77) reaches 8.03% of the total events.

**Table 5.** Parameters and results of clustering iterations for the engine speed signal of $NO_X$ events.

|  |  | **Loop 1** | **Loop 2** | **Loop 3** | **Loop 4** |
|---|---|---|---|---|---|
| Number of events | - | 959 | 448 | 203 | 159 |
| $C_{minSize}$ | - | 4 | 3 | 5 | 3 |
| $\varepsilon_{min}$ | % | 1 | 1 | 1 | 1 |
| Resulting clusters | - | 47 | 15 | 4 | 3 |
| Detected outliers | - | 448 | 203 | 159 | 77 |
| *DBCV* | - | 0.589 | 0.495 | 0.303 | 0.403 |
| Average Silhouette Score | - | 0.26 | 0.23 | 0.1 | 0.13 |

The *DBCV* value shows an acceptable level, although being lower than for the verification analysis, for all iterations and is lowest at 0.303 for the third run. The average Silhouette Scores show rather low values. Given the high variance in the signal profiles, this is mostly due to individual events that do not fit perfectly into a cluster and tend to be interpreted as a transition to a neighboring cluster. The courses of the Silhouette Scores are exemplarily shown for the result cluster set of Loop 2 and Loop 3 in Figure 8.

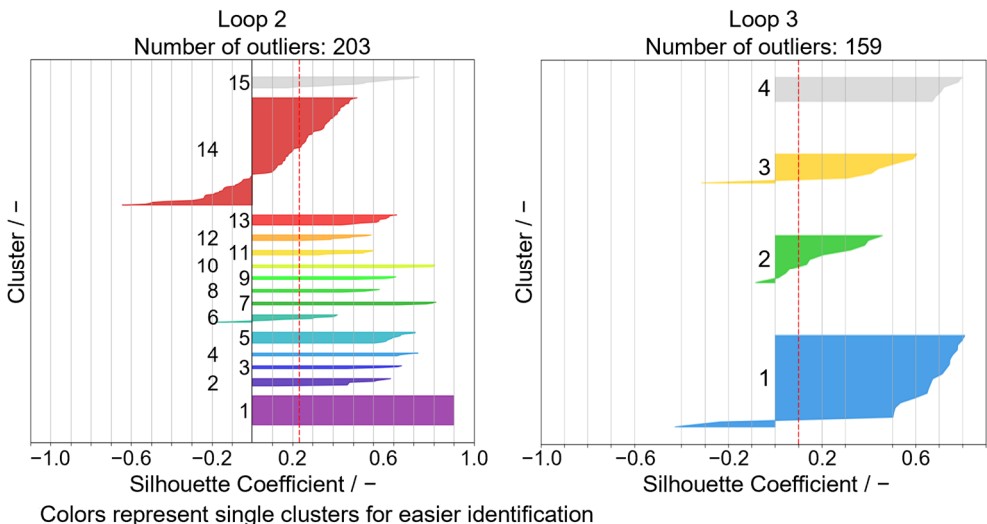

**Figure 8.** Silhouette Score of cluster results for Loop 2 and Loop 3.

Finally, the sets of individual clusters from the iterative loops are merged. The assigned clusters of outliers are inserted into the results of the previous loop successively. In this way, only the cluster assignment of classified outliers is corrected for each iteration. The 69 raw clusters are judged manually for the final interpretation. Similar clusters are combined based on engineering judgement. Merging neighboring clusters with numerical differences but similar system behavior reduces the number to 24, resulting in 63 events as outliers. The result is shown in Figure 9.

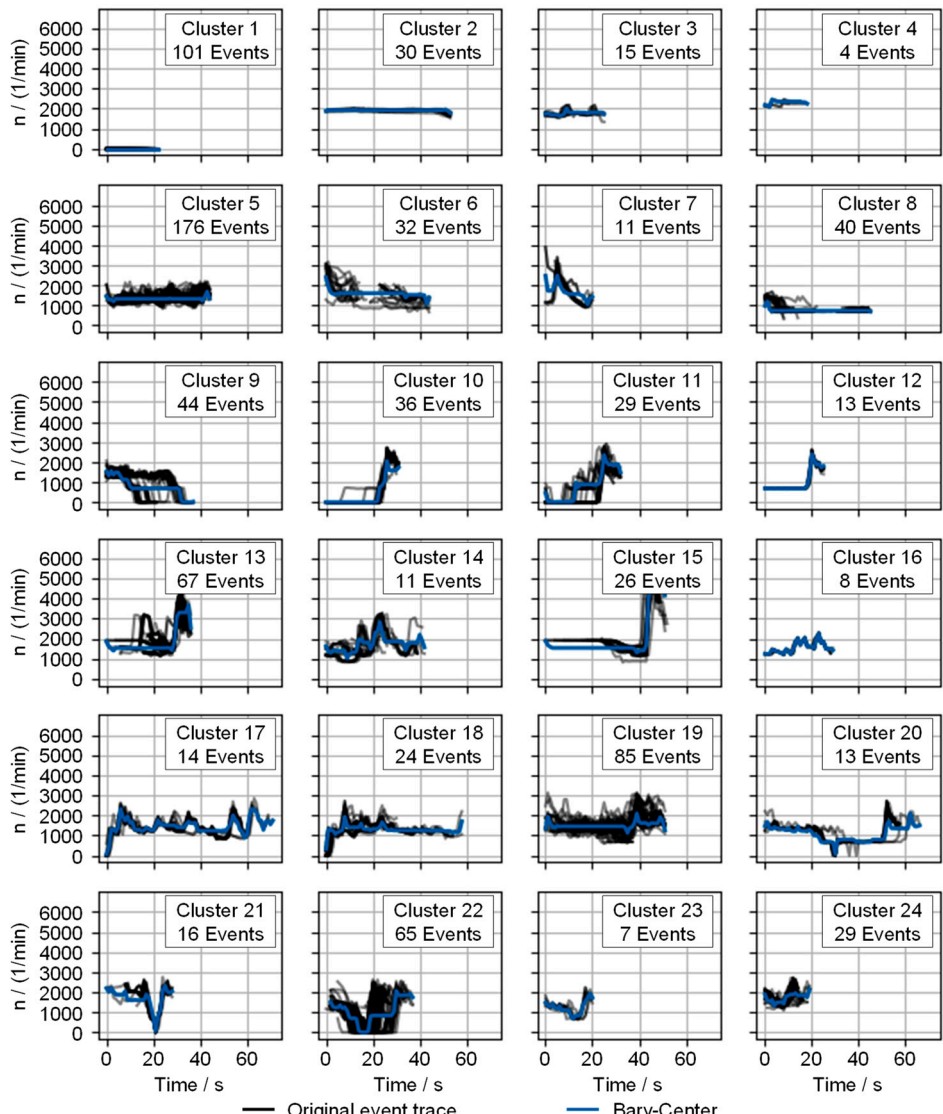

**Figure 9.** Overview of the final cluster definition for the engine speed signal of NO$_X$ events.

The representing trace of a cluster is again expressed by the bary-center in blue. The original event traces are shown with black lines. Various phenomena are exhibited by the characteristics of the clusters. Cluster 1 contains events with an inactive combustion engine (cf. cluster 3 from the HDBSCAN verification in Section 3.2). Clusters 2–4 depict continuous engine speed progressions. Due to the varying levels and durations of the events, different clusters are formed. Cluster 5 is characterized by slightly oscillating speed profiles between $n = 1000\ 1/\text{min}$ and $n = 2000\ 1/\text{min}$. The subsequent clusters 6–9 display profiles with an initial braking phase.

Clusters 10 and 11 display engine start events and could be consolidated. Clusters 12 through 15 exhibit distinct acceleration processes. Specifically, cluster 13 contains single accelerations (Figure 10), whereas cluster 14 centers on repeated accelerations. The similarity in the representation of these stems from the varying duration of events, which is evident for cluster 13 in Figure 10.

The other clusters are formed by phases of different dynamics with slight or strong fluctuation. Phases with included standing phases are evident. While cluster 21 contains only short standstill times (mostly $\Delta t = 1$ s to $\Delta t = 2$ s), the standstill times in cluster 22 vary.

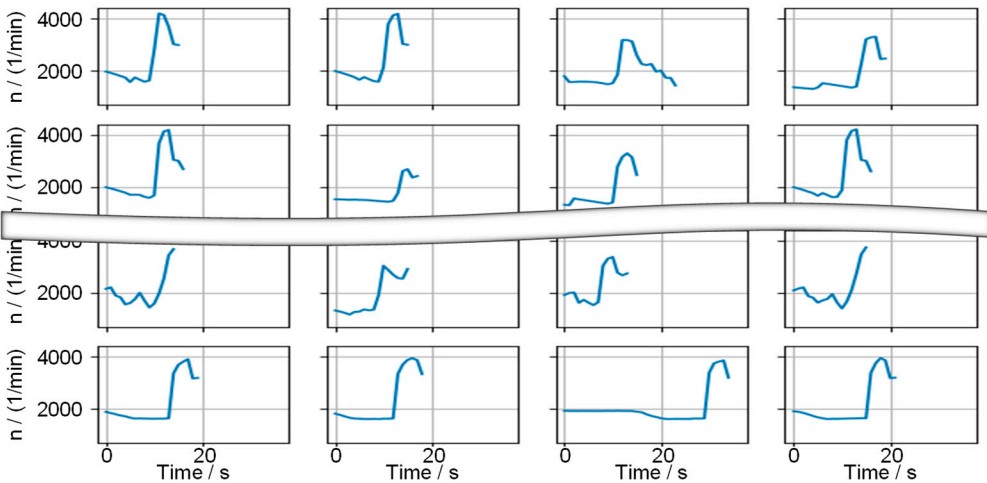

**Figure 10.** Extract of cluster 13's event profiles.

## 4. Discussion

The discussion of the presented results is separated into sections including the evaluation criteria, the data pre-processing procedure, and the application of the clustering method to the emission measurements.

### 4.1. Evaluation Criteria and HDBSCAN Validation

The *ARI* correlations between the HDBSCAN results in Section 3.2 and the defined reference clusters show a predominantly good quality of the automated results (Table 4). At the same time, the number of detected outliers is low. This supports the use of HDBSCAN. While the possibility of detecting outliers is preferred, the reference dataset does not include any outliers. In the visual validation, the specified outliers show reasonable traces to accept them as such. Especially for the resolution of clusters, the determination of the optimal size is not perfect, and deviations from the reference distribution are expected and accepted. The results show that the automated clustering tends to over-determine clusters. Except for the clusters for speed, engine torque, relative fuel mass, and optimal ignition angle, more clusters are formed with automated clustering (Tables 3 and 4). For the accelerator pedal position, automated clustering divides the data into one fewer cluster than manual clustering. A division into smaller clusters is more practicable than a few large clusters. A manual merging of clusters can be realized easily and rapidly with a graphical evaluation. A manual splitting of events, however, is more difficult.

The evaluation of the comparison of automatic clustering to the manually created reference clusters in Section 3.2 shows an overall good correlation between *ARI* and *DBCV*. In some cases, as, e.g., for the relative air charge or the relative fuel mass, the indications from *ARI* and *DBCV* do not align. As an external reference-based measure, the *ARI* results are preferred. Except for the fuel cut-off flag and the temperature of the catalytic converter, the *ARI* values show a higher rating of the cluster quality than the internal density based on *DBCV*. In daily use, the assessment of quality using *ARI* is not practical. However, the results of the cluster assessment show an overall good correlation between *DBCV* and *ARI* (Table 4). As an intensive criterion, *DBCV* does not require further processing or preparation of the data. Thus, *DBCV* is preferred for automated application in calibration.

### 4.2. Pre-Processing of Data

The distance matrix is created using the EDTW method on the raw signals. During the application of the method, the risk of overfitting data toward each other is seen. For the high variation in the compared data, the key feature of DTW—allowing to compare signals of different lengths and to correct offsets and slight differences—is useful. To overcome the risk of falsifying data using DTW, the complexity estimate correction is

useful. Although, the influence of DTW cannot always be predicted. A visual correction of assigned events to clusters still must be carried out and can help to identify weak spots in the DTW comparison. While using only one signal at a time for the clustering approach, the validation of DTW falsification is possible. Overcoming the risks of DTW in feature-based clustering approaches might be possible when not relying on a time-series-based distance calculation directly. Approaches based on Fourier transformation or signal feature-based multi-dimensional clustering methods are yet to be further analyzed.

The signals are not normalized in advance. An investigation with normalized values on the cluster quality is still pending. The normalization significantly restricts the range of values of the event-to-event distances. This may lead to further simplification in the design of the cluster parameters. In the current non-normalized approach, the distances are strongly dependent on the magnitude of the respective signal. Thus, the findings of distance magnitudes and thresholds are not transferable across the signals without normalization. An impact on the amount of outlier classification for the complete dataset is yet to be analyzed.

*4.3. Judgement of HDBSCAN Application for Emission Calibration Purposes*

The results show that HDBSCAN provides an automated method for categorizing time signals of different characteristics. In emission calibration, this method allows to significantly reduce the amount of data to be analyzed and offers the potential to quantify and weigh weak points. The automated event detection condenses the total test data into only the critical parts. In the project used, 78 emission measurements with a total duration of 255,928 s are reduced to a quantity of 959 events relevant to $NO_X$ emissions. These 959 events have a total duration of 21,827 s. The automatic clustering, based on the engine speed signal in Section 3.3, with a total of 69 raw clusters shows the tendency of over-classification and the formation of micro-clusters. A manual visual correction of the clusters results in 24 final clusters. Here, automatically assigned clusters are only merged, and splitting an existing cluster is not required. Although this step required manual engineering effort, this procedure is suggested. Variations in the HDBSCAN settings show that the formation of rather large clusters is not desirable.

While automated clustering (distance matrix calculation and HDBSCAN execution) of the given dataset requires only a couple minutes (excluding the event detection and data preparation), manually merging the clusters on a visual base requires around one hour for an engineer who is familiar with the data and the procedure. While this significantly increases the time-based effort, it allows to manually control the sensitivity of the data assignment for later analyses. The required manual data analysis for categorization is reduced to the evaluation of the overall cluster plots.

Depending on the selected signal, the categorization of events can lead directly to relevant weak spots. However, for a detailed weak spot analysis, it is necessary to consider the influence of a multitude of signals. This can be performed either by evaluating similar combinations of groups in different cluster sets based on different signals or by applying the clustering process to a multitude of signals at once. Both concepts are currently under investigation and will be discussed in subsequent publications.

Although the clustering of a single signal describing a driving maneuver (e.g., vehicle speed, engine speed, or engine torque) can provide a first indication of the general nature of weak spots, automatic clustering enables the prioritization of actions to optimize the system behavior within the identified events by quantifying their impact.

Figure 11 shows the distribution of distance-specific $NO_X$ emission intensity of the events in the previously presented clusters. While the expected intensity of each cluster is below 0.2 g/km for most clusters, clusters 2, 10, 11, 12, 13, 17, and 18 show rather high intensities. Clusters 17 and 18 are the clusters that summarize statistically the most intense events. Cluster 10 and cluster 11 contain the highest overall events considering the outliers in the boxplot intensity distribution.

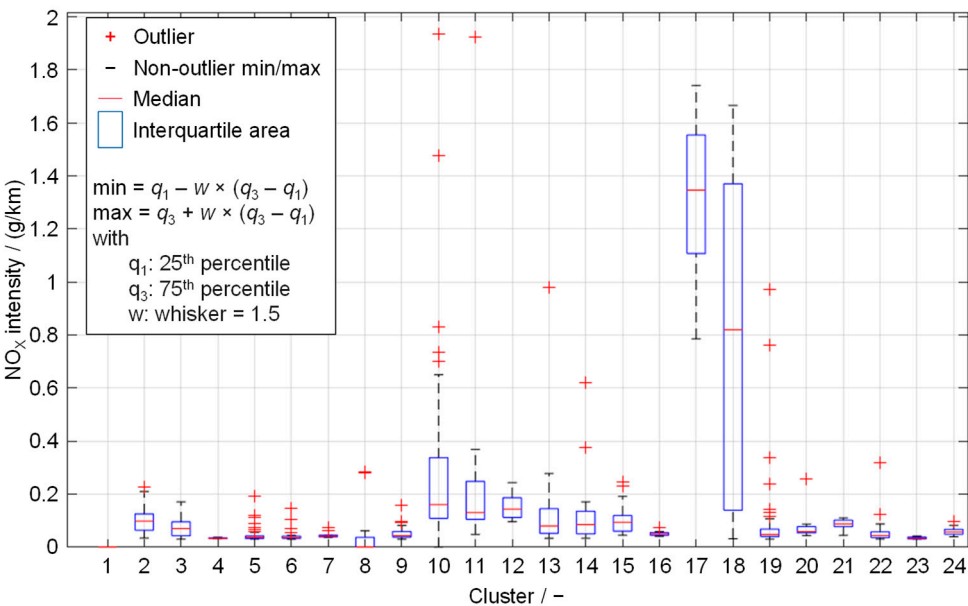

**Figure 11.** Boxplot of distance-specific NO$_X$ emission intensity distribution.

Similar analyses can be performed using different factors. In addition to the specific emission intensity distributions, information such as cumulative intensity (multiplied by cluster size), distance traveled, duration, appearance in the measurement after engine start, EATS temperature, high voltage battery SOC, etc., can be used to further judge the root causes and correlations of clusters. However, when considering the emission signal or absolute intensity, the origin of the measurement should be considered. Measured emission traces can vary in shape depending on the measurement system used (e.g., emission chassis dynamometer with constant volume sampling measurement vs. on-road PEMS measurement). Furthermore, the intensity of similar events varies with different aging conditions of the EATS.

A further detailed analysis of the events presented in this study overreaches the scope of this presentation. The presented initial categorization enables weak spots to be quantified. Thus, the total impact of a weak spot can be calculated and evaluated by the total number of events and their intensity by the cluster. This enables both a prioritization in addressing the identified weak spots and a quantitative comparison of the development of these over the ECU datasets during a project. By creating and comparing cluster sets at different stages of the calibration process, it is possible to assess whether the actions taken are simply shifting the weak spots to another issue or are optimizing the overall system behavior.

## 5. Conclusions

With the shift toward evaluating emissions in vehicle development in real-world conditions introduced by EU6d and further planned in EU7, validation methods and innovative test benches are becoming increasingly important. However, to efficiently exploit resulting potentials, new approaches in data evaluation are required. To efficiently optimize a system's calibration, weak spots must not only be identified, but patterns also need to be identified.

This paper explores the potential of clustering emission data for quantifying weak spots. Using automatic event detection, complete measurements are summarized to include only the relevant parts necessary for the calibration process. This automatic process saves time and improves efficiency.

Dynamic time warping is used to construct a distance matrix based on time series comparisons to automate the processing and comparison of events of different durations. DTW has a high potential for reducing the impact of minor differences and offsets in

two signals. However, there is also a risk of overfitting. In an unlimited DTW application, the complexity estimate function can account for various signal shapes and durations, factoring in the distance based on the initial signal's difference in complexity and duration.

An initial evaluation on the usability of HDBSCAN for vehicle calibration data is conducted by manually creating reference cluster sets for 12 different signals. The automatic HDBSCAN application is then compared to the same extract of events, resulting in an overall strong correlation between manual clustering and automatic clustering with an $ARI = 0.74$. Already here, the algorithm displays a tendency to create multiple smaller clusters.

The algorithm applied to the entire dataset of 959 events detected a significant number of outliers (nearly 50% of the total data) during clustering solely based on the engine speed signal. To address this issue, a procedure is introduced that involves re-clustering the identified outliers in a separate run. This is iterated until 8.03% of outliers are reached in the used dataset.

The application of the complete dataset confirms the tendency to identify a high number of small clusters. HDBSCAN's iterative execution on outliers yields a total of 69 clusters. From this, 24 clusters of engine speed are summarized manually, while the remaining 63 events are considered outliers in the final distribution. While automatic clustering requires under 10 min, the manual merging of clusters based on a visual analysis of the given amount of clusters and events demands significantly more time, with a one-hour investment. Nevertheless, the manual merging of clusters increased the number of clusters, allowing for greater freedom in terms of the application-specific sensitivity of the results.

The presented application of clustering is performed using only a single signal to decide on the categorization of events. While more complex than an initial validation and presentation of the approach, a detailed root cause analysis requires a larger number of signals to be considered. Such applications are currently under investigation and will be presented in subsequent publications. However, even single-signal clustering offers a high benefit for the data analysis itself, as driving maneuvers (most expressed by the speed signal) can already be clustered to obtain a first impression of the available data. In addition, it can support the current manual analysis by being able to cluster data based on an already existing manual assumption. In this way, a quantification of the actual impact as well as further correlation studies on vehicle or environmental effects can be performed.

While the application is focused on emission calibration, the design of the proposed data analysis method can be flexibly applied to any use case. For example, if the event definition is not critical in terms of emission intensity but for a hybrid operating strategy, electrical energy consumption, or issues such as component protection and derating strategies on electrified vehicles in the future.

**Author Contributions:** Conceptualization, S.K.; methodology, S.K. and G.T.; software, S.K. and G.T.; validation, S.K. and G.T.; formal analysis, S.K. and G.T.; investigation, S.K. and G.T.; resources, S.K., M.G., M.N. and S.P.; data curation, S.K. and J.C.; writing—original draft preparation, S.K.; writing—review and editing, J.C., G.T., M.D., F.D., M.G., M.N. and S.P.; visualization, S.K. and G.T.; supervision, M.G., M.N. and S.P.; project administration, S.K.; funding acquisition, S.P. All authors have read and agreed to the published version of the manuscript.

**Funding:** The presented research was carried out at the Center for Mobile Propulsion (CMP) of RWTH Aachen University, funded by the German Science Council "Wissenschaftsrat" (WR) and the German Research Foundation "Deutsche Forschungsgemeinschaft" (DFG).

**Institutional Review Board Statement:** Not applicable.

**Informed Consent Statement:** Not applicable.

**Data Availability Statement:** The data presented in this study are available upon request from the corresponding author. The data are not publicly available due to the complexity of the analysis which needs guidance for reproduction and confidentiality of root measurements.

**Conflicts of Interest:** The authors declare no conflicts of interest. The funders had no role in the design of this study; in the collection, analyses, or interpretation of data; in the writing of this manuscript, or in the decision to publish the results.

## List of Abbreviations

| | |
|---|---|
| ARI | Adjusted Rand Index |
| AT | automatic transmission |
| AWD | all-wheel drive |
| CE | complexity estimate |
| cf. | confer |
| CF | complexity factor |
| CO | carbon monoxide |
| DBCV | Density-Based Cluster Validation |
| DoE | Design-of-Experiments |
| DTW | dynamic time warping |
| e.g. | for example |
| EATS | exhaust aftertreatment system |
| ECU | engine control unit |
| EDTW | complexity estimate dynamic time warping |
| EiL | Engine-in-the-Loop |
| FN | False Negative |
| FP | False Positive |
| GPF | gasoline particulate filter |
| HC | hydrocarbons |
| HDBSCAN | Hierarchical Density-based Spatial Clustering of Applications with Noise |
| HiL | Hardware-in-the-Loop |
| MiL | Model-in-the-Loop |
| NOX | nitrogen oxides |
| PEMS | portable emission measurement system |
| PiL | Powertrain-in-the-Loop |
| RDE | real driving emissions |
| RI | Rand Index |
| SOC | state of charge |
| TN | True Negative |
| TP | True Positive |
| TWC | three-way catalytic converter |
| ViL | Vehicle-in-the-Loop |
| WLTC | Worldwide Harmonized Light Vehicles Test Cycle |
| XiL | X-in-the-Loop |

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
