# Peer review of "Applying Density-Based Clustering for the Analysis of Emission Events in Real Driving Emissions Calibration"

_futuretransp, doi:10.3390/futuretransp4010004_

Round 1
Reviewer 1 Report
Comments and Suggestions for Authors
Overall, I found the manuscript to be an interesting read. However, I would Overall, I found the manuscript to be an interesting read. However, I would like to suggest a major revision to address several key areas that require improvement. In the following sections, I will outline these improvements and provide specific feedback to help enhance the quality and clarity of the manuscript.
1. Clarity and Structure:
- Enhance manuscript structure with clearer sections, headings, and subheadings.
- Improve writing clarity, ensuring concise, well-structured sentences.
- Define technical terms and fix abbreviations and cross-references.
2. Introduction:
- Provide a comprehensive overview of the problem, research objectives, and significance.
- Expand context beyond Euro 6 and Euro 7 emissions standards to include current electrification trends.
3. Methodology:
- Simplify the explanation of the comprehensive methodology due to its length.
4. Data:
- Explain the initial dataset in more detail, including how you end with the 959 events.
- Address the impact of DTW on data alignment and suggest ways to mitigate alignment-related distortions.
- Conduct a deeper analysis of clustered data, emphasizing its relevance to the study's objectives.
5. Limitations:
- Elaborate on how acknowledged limitations may impact study outcomes and interpretations.
- Provide context on the significance of these limitations.
6. Interpretation and Discussion:
- Expand the discussion section to offer a more profound interpretation of findings.
- Emphasize real-world implications for emissions control.
7. Conclusion:
- Ensure the conclusion effectively summarizes key findings, their significance, and potential future research directions.
8. References:
- Consider removing some references and adding relevant ones about EU or global pollution and emissions targets and legislation.
Comments on the Quality of English LanguageGood level, however proofreading is needed to address several mistakes.
Reviewer 2 Report
Comments and Suggestions for Authors
Real driving emissions (RDE) legislation poses a significant challenge to the vehicle calibration process, with quasi-infinite combinations of test scenarios resulting in large amounts of data that calibration engineers must analyze. The paper demonstrates the application of the HDBSCAN clustering procedure to the emissions calibration process, which enables the automatic identification, classification, and prioritization of calibration defects. Still, the paper has some questionable points that need minor revision.
1. Line 124: Isn't there an error, ‘Figure 1’?
2. Line 146: What is ‘a specified emission intensity threshold’?
3. Lines 158‒161: The paper uses a Dynamic Time Warping (DTW) approach to consider the different durations. But there are problems with this approach. For example, an event of rapid acceleration may be in the same category as an event of slow acceleration, but in reality, the two events have significantly different effects on vehicle emissions.
4. Lines 174‒268: The paper devotes a great deal of space to the HDBSCAN clustering program. If the authors have not improved on this standard algorithm, the introduction of the algorithm could be outlined. In addition, clustering algorithms are very rich (e.g. systematic clustering, ordered sample clustering, dynamic clustering, fuzzy clustering, graph theoretic clustering). Why was the HDBSCAN algorithm chosen, and what makes the HDBSCAN algorithm better suited for the processing of calibration data?
5. Lines 271‒276: Data sources need to be described in more detail, e.g., what type of the test vehicle? Tested under laboratory conditions or on actual roads? Or is the data available for both laboratory and road test conditions? What are the 78 measurements (test procedures)? How were these 959 events of NOx critical sequences detected?
6. Lines 279‒280: The cluster methodology based on only one signal is flawed for the RDE calibration application. Why is the method only with one signal?
7. Line 427: The principle of re-clustering needs to be explained.
8. Line 449: Figure 8 shows the final 24 clusters for the engine speed signal of NOx events. But do the NOx emissions from these clusters exhibit distinguishable clustering characteristics?
9. Line 452: Cluster 1 includes events with inactive combustion engine. Could it be that non-starting engines produce critical NOx emissions?
10. Lines 520‒530: This text feels a bit repetitive with that in the introduction section.
11. In conclusion, the research is based on clustering for a single signal only, whereas cluster analysis for combining multiple signals can be more valuable for emission calibration. The fact that the paper is based on clustering for a single signal should be explained in the abstract or the last paragraph of the introduction section to make it easier for the reader to understand.
Reviewer 3 Report
Comments and Suggestions for Authors
The authors have written a comprehensive article on “Applying Density-Based Clustering for Analysis of Emission Events in RDE Calibration”. They have discussed a clustering method in RDE emissions. However, the following comments and queries must be addressed for further processing:
1. The flow connectivity is missing in the abstract section. Add existing challenges, proper methodology, findings, and conclusive results. Moreover, add some quantitative data in the abstract section.
2. The English language applied is rich from the viewpoint of used vocabulary but rather hardly comprehensible. Thus, the entire manuscript ought to be thoroughly elaborated, perhaps once again, by a properly skilled in this field English language expert.
3. There are lumpy many references in the introduction section. Each and every reference should be cited properly.
4. The introduction part is not properly organized. Moreover, the research gap is not properly identified and explained in the introduction section.
5. Regarding RDE emissions there are some papers that are still missing in literature. The authors can take help from these papers.
https://doi.org/10.1016/j.apr.2022.101597
https://doi.org/10.1016/j.fuel.2021.121642
6. The main objective of the paper must be written in a clearer and more concise way at the end of the introduction section.
7. The novelty of the work must be clearly addressed and discussed, compare your research with existing research findings and highlight novelty.
8. It's crucial that all figures and their related content are aligned properly with the text. Additionally, the quality of the figures needs improvement to ensure clarity. (e.g Figures 1, 6 and 8).
9. All of methods used in this paper are derived from other paper, the authors seem only put them all together. While, what are the innovation points of this paper? Or can the authors prove the reasonability of proposed method in theory?
10. The figure 6 should be redrawn, it is too illegible.
11. The drafting style of the paper is not good and not properly organized. Please go through each paragraph and correct it.
Comments on the Quality of English LanguageExtensive editing of English language required.
Reviewer 4 Report
Comments and Suggestions for Authors
The manuscript presents a clustering methology for RDE driving cycles to reduce the calibration effort of ECUs. The methodology is properly introduced, and the results are discussed in detail. Drawbacks of the method are identified and possible solutions are suggested.
There is only one minor issue with a reference on page 3, line 124.
Round 2
Reviewer 1 Report
Comments and Suggestions for Authors
Author's have taken into account this reviewer's comments and significantly improved the flow and quality of this manuscript. Would reccomend this manuscript for publication after proofreading.
Comments on the Quality of English LanguageThere are minor mistakes, style consistency should be improved
Reviewer 3 Report
Comments and Suggestions for Authors
The responses provided by the authors in relation to my previous comments have been thoroughly addressed, and I am pleased to recommend that this paper be considered for further publication in Future Transportation, pending the completion of the prerequisite settings specific to the journal.